# Physicochemical Perturbation Increases Nitrous Oxide Production from Denitrification in Soils and Sediments

Nathaniel B. Weston[1], Cynthia Troy[1], Patrick J. Kearns[2,3], Jennifer L. Bowen[2], William Porubsky[4], Christelle Hyacinthe[5], Christof Meile[5], Philippe Van Cappellen[6] and Samantha B. Joye[5]

[1] Department of Geography and the Environment, Villanova University, Villanova, PA 19085, USA
[2] Department of Marine and Environmental Sciences, Northeastern University, Boston, MA 02115, USA
[3] Current address: Department of Biology, University of Massachusetts Boston, Boston, MA 02125, USA
[4] Algenol Biofuels, Fort Meyers, FL, 33912, USA
[5] Department of Marine Sciences, University of Georgia, Athens, GA 30602, USA
[6] Water Institute, Department of Earth and Environmental Sciences, University of Waterloo, Waterloo, ON, Canada

*Correspondence to*: Nathaniel B. Weston (nathaniel.weston@villanova.edu)

**Abstract.** Atmospheric concentrations of nitrous oxide ($N_2O$), a potent greenhouse gas that is also responsible for significant stratospheric ozone depletion, have increased in response to intensified use of agricultural fertilizers and other human activities that have accelerated nitrogen cycling processes. Microbial denitrification in soils and sediments is a major source of $N_2O$, produced as an intermediate during the reduction of oxidized forms of nitrogen to dinitrogen gas ($N_2$). Substrate availability (nitrate and organic matter) and environmental factors such as oxygen levels, temperature, moisture, and pH influence rates of denitrification and $N_2O$ production. Here we describe the role of physicochemical perturbation (defined here as a change from the ambient environmental conditions) on denitrification and $N_2O$ production. Changes in salinity, temperature, moisture, pH, and zinc in agricultural soils induced a short-term perturbation response characterized by lower rates of total denitrification and higher rates of net $N_2O$ production. The $N_2O$ to total denitrification ratio ($N_2O$:DNF) increased strongly with physicochemical perturbation. A salinity press experiment on tidal freshwater marsh soils revealed that increased $N_2O$ production was likely driven by transcriptional inhibition of the nitrous oxide reductase (nos) gene, and that the microbial community adapted to altered salinity over a relatively short (within one month) timeframe. Perturbation appeared to confer resilience to subsequent disturbance, and denitrifiers from an environment without salinity fluctuations (tidal freshwater estuarine sediments) demonstrated a stronger $N_2O$ perturbation response than denitrifiers from environments with more variable salinity (oligohaline and mesohaline estuarine sediments), suggesting that the denitrifying community from physicochemically stable environments may have a stronger perturbation response. These findings provide a framework for improving our understanding of the dynamic nature of $N_2O$ production in soils and sediments, in which changes in physical and/or chemical conditions initiate a short-term perturbation response that promotes $N_2O$ production that moderates over time and with subsequent physicochemical perturbation.

## 1 Introduction

Human activities continue to accelerate the global nitrogen (N) cycle through the industrial fixation of dinitrogen gas ($N_2$) for use as agricultural fertilizer, increased cultivation of N-fixing crops, and combustion of fossil fuels (Galloway et al., 2004). As a result, the availability of reactive N continues to increase in terrestrial and aquatic systems worldwide. Because many ecosystems are N limited (Vitousek & Howarth, 1991), increased levels of reactive N in the biosphere can have deleterious impacts, including the eutrophication of inland and coastal waters (Nixon, 2009). Denitrification is an anaerobic pathway of

microbial respiration that removes reactive N through the reduction of inorganic nitrogen [nitrate ($NO_3^-$) or nitrite ($NO_2^-$)] to unreactive dinitrogen gas ($N_2$; Payne, 1973; Knowles, 1982; Seitzinger, 1988). The complete reduction of $NO_3^-$ to $N_2$ occurs in several steps that require the reduction of the intermediate gases nitric oxide (NO) and nitrous oxide ($N_2O$) and is

accomplished by a series of enzymatic reactions catalyzed by $NO_3^-$ reductase (Nar), $NO_2^-$ reductase (Nir), NO reductase (Nor), and $N_2O$ reductase (Nos; Knowles, 1982). $N_2O$ is produced transiently during denitrification, and some $N_2O$ escapes reduction and is emitted from zones of active denitrification to overlying waters and/or the atmosphere (Seitzinger, 1988). The increase in global reactive N fuels greater rates of denitrification, resulting in increased emissions of $N_2O$ from soils, sediments, and waters (Denman et al., 2007; Beaulieu et al., 2011; Tian et al., 2020). $N_2O$ is also produced through fungal denitrification

(Maeda et al., 2015), microbial nitrification (Davidson et al., 1986), chemodenitrification (abiotic denitrification; Grabb et al., 2017; Robinson et al. 2021), and, possibly, dissimilatory nitrate reduction to ammonia (Butterbach-Bahl et al., 2013). The contribution of these processes to global $N_2O$ budgets is less clear, but in many instances where direct comparisons have been made, bacterial denitrification is often the dominant $N_2O$ source from soils and sediments (Mathieu et al., 2006; Vilain et al., 2014; Hu et al., 2015). More recently, however, (Bahram et al., 2022) found that Archaeal nitrifiers may play a more important

role in $N_2O$ production in soils than previously recognized. In the troposphere, $N_2O$ is a potent greenhouse gas with a global warming potential 298 times that of carbon dioxide over a 100-year timeframe (Forster et al., 2007). Concentrations of $N_2O$ in the atmosphere have risen by more than 18% with an estimated increase of roughly 0.26 % per year from 1980 through 2005 (Forster et al., 2007). In addition, $N_2O$ is currently the single most important ozone-depleting atmospheric trace gas and is expected to remain so throughout the 21st century (Ravishankara et al., 2009). Given the potency of $N_2O$ as a greenhouse

gas and ozone depleting substance, a better understanding of $N_2O$ production dynamics in the geosphere is needed (Wuebbles, 2009).

Despite the importance of $N_2O$ to climate change and stratospheric ozone dynamics, the factors that regulate net $N_2O$ production from soils and sediments during denitrification (DNF; defined here as the sum of $N_2O$ and $N_2$ production) remain unclear, and we do not yet know why the ratio of $N_2O$ production to total denitrification ($N_2O$:DNF) varies in denitrifying

environments. Denitrification rates are spatially and temporally heterogeneous in soils and sediments, resulting in 'hotspots' and 'hot moments' of activity (McClain et al., 2003; Groffman et al., 2009). Likewise, $N_2O$ emissions from soils vary considerably over space and time, and our ability to predict this variation is limited (Huang et al., 2011; Henault et al., 2012; Harrison-Kirk et al., 2013; Weitzman et al., 2021). Several environmental variables impact rates of denitrification and $N_2O$ emissions, including the availability of substrates ($NO_3^-$, labile organic matter, and other electron donors such as ferrous iron).

In general, the proportion of $N_2O$ released from soils increases with increasing $NO_3^-$ availability (Firestone et al., 1980; Barnard et al., 2005; Bao et al., 2012) and since denitrification is an anaerobic respiration process, it can be sensitive to oxygen ($O_2$) concentrations and soil moisture levels, which affects $O_2$ diffusion into soils (Firestone et al., 1980; Seitzinger, 1988; Conrad, 1996; Wang et al., 2023). Although rates of denitrification generally decline as oxygen concentrations increase (Knowles, 1982; Rosamond et al., 2012), the $N_2O$:DNF ratio can increase with higher $O_2$ availability (Firestone et al., 1980; Betlach &

Tiedje, 1981; Burgin & Groffman, 2012).

In addition to substrate availability and $O_2$, other soil/sediment physicochemical factors can influence rates of denitrification and $N_2O$ production. Soil pH exerts a potential control on rates of denitrification and $N_2O$:DNF ratios (Firestone et al., 1980; Weslien et al., 2009; Baggs et al., 2010). Typically, under more acidic conditions rates of denitrification are lower and the $N_2O$:DNF ratio is higher (van den Heuvel et al., 2011; Raut et al., 2012). Liu et al. (2014) suggest that posttranscriptional

inhibition of the nitrous oxide reductase enzyme under lower pH conditions was responsible for the greater relative $N_2O$ production. Similarly, increasing concentrations of heavy metals inhibits the reduction of $N_2O$, leading to higher $N_2O$ fluxes (Magalhaes et al., 2007; Ruyters et al., 2010). Temperature (Seitzinger, 1988; Larsen et al., 2011; Billings & Tiemann, 2014), hydrogen sulfide (Porubsky et al., 2009), and salinity (Giblin et al., 2010; Teixeira et al., 2013) can also exert control on

denitrification and N$_2$O production. While physicochemical conditions can clearly influence denitrification, our understanding of how environmental controls impact denitrification coupled with N$_2$O production remains limited (Butterbach-Bahl et al., 2013), and the resilience of microbial communities to changes in physicochemical conditions is not straightforward (Griffiths & Philippot, 2013). We addressed this knowledge gap by investigating whether pulse and press disturbances (Bender et al., 1984) that arise from changing physicochemical conditions elicit a perturbation response from the denitrifying community. We define perturbation as a deviation from ambient physicochemical conditions encountered by the denitrifying microbial community in soils or sediments. We explore how perturbation alters rates of denitrification, N$_2$O production, the N$_2$O:DNF production ratio, and changes in the gene expression of the nitrite reductase (*nirS*) and nitrous oxide reductase (*nosZ*) genes that code for key enzymes that mediate N$_2$O production and consumption.

## 2. Methods

Several experiments were conducted to evaluate environmental controls on denitrification (defined as N$_2$ + N$_2$O production), N$_2$O production, and the N$_2$O to total denitrification ratio (N$_2$O:DNF). We conducted three discrete experiments (Table 1) that addressed 1) the short-term perturbation response induced by manipulation of the physicochemical (salinity, temperature, pH, soil moisture, and zinc toxicity) status of agricultural soils, 2) the short-term response of denitrifying communities from environments experiencing a range in one parameter (salinity in estuarine sediments) to changes in that parameter, and 3) the long-term response (changes in process rates and gene expression) of the denitrifying community to a change in a single parameter (salinity in estuarine sediments). We elected to focus on changes in salinity in the 2$^{nd}$ and 3$^{rd}$ experiments because it is a parameter that changes over daily (tidal) and seasonal timescales in estuarine environments and is therefore ecologically relevant, it is a parameter that will be altered in some environments in response to climate change, and it is relatively easy to manipulate and to measure. We did not have the resources to investigate additional physicochemical parameters beyond the 1$^{st}$ experiment. In all experiments, soils/sediments were incubated in oxygen-free, gas-tight headspace vials and the production of N$_2$O was measured with and without acetylene (Balderston et al., 1976; Yoshinari & Knowles, 1976; Groffman et al., 2006). N$_2$O production rates without acetylene reflect N$_2$O produced by the microbial community. Acetylene inhibits N$_2$O reductase and thus blocks the final step in the denitrification process, resulting in the buildup of N$_2$O rather than N$_2$ (Balderston et al., 1976; Yoshinari & Knowles, 1976). N$_2$O production rates with acetylene therefore reflect the total rate of denitrification (N$_2$ + N$_2$O; DNF). Headspace gas samples were taken at several (typically 3 to 4 times) during the incubation (see Appendix for example) and rates of denitrification and N$_2$O production were calculated from the linear increase in N$_2$O over time. Incubation times were relatively short (typically <12 hr) to avoid changes in denitrifier population, longer-term adaptation of the denitrifying community to changes in physicochemical disturbance, and changes in substrate concentrations. In instances where nonlinearity in the production of N$_2$O was observed due to these (or other) factors, the data from later timepoints was not used and only the linear portion of the timecourse incubations were used to calculate N$_2$O production. Rates are reported as μmol N$_2$ + N$_2$O (DNF) or N$_2$O per gram fresh soil/sediment per day (μmol g$^{-1}$ d$^{-1}$). The ratio of N$_2$O produced to total denitrification (N$_2$O:DNF) was calculated on a per mole basis.

### 2.1 Agricultural Soils – Salinity, pH, Zinc, Temperature and Moisture Pulse Perturbations

Experiments were conducted on agricultural soil samples collected from two sites in July 2011, one farmed conventionally (40.07464 N, 76.212008 W) and one farmed using organic practices (40.069779 N, 76.238079 W), in Lancaster, PA. Surface soils (0-2 cm) were collected from each site. The temperature of the surface soils (37 °C) was measured at the time of collection. The soils were returned to the laboratory, homogenized, and visible roots were removed. The soil water content (0.48 g g$^{-1}$ for the conventional soil and 0.46 g g$^{-1}$ for the organic soil) was determined by the loss of weight upon drying at 80 °C for 48

hours, and soil pH (7.37 in the conventional soil and 7.09 in the organic soil) was measured with a pH probe after mixing 20 g of soil with 25 ml deionized water. For each perturbation assay, approximately 20 g of soil was placed into a 410 mL headspace jar and treatments with varying salinity, pH, zinc, temperature, and moisture were achieved to evaluate changes in $N_2O$ production and denitrification rates (Table 1). All experiments except for the moisture treatment received 10 mL of water, and all treatments were amended with 1 mM $NO_3^-$ and 2 mM glucose.

A series of jars were amended to achieve various salinities (0, 1, 3, 5, 10, and 30 g $kg^{-1}$) using an artificial saltwater solution (350 mM NaCl, 45.5 mM $MgCl_2$, 24.2 mM $Na_2SO_4$, 8.9 mM $CaCl_2$, 2 mM $NaHCO_3$, and 0.5 mM KCl for salinity of 30 g $kg^{-1}$ and diluted as appropriate with deionized water for other salinities). Similarly, a series of jars were amended to obtain various zinc concentrations (0, 0.05, 0.1, 0.25, 0.5 and 1.0 g Zn $L^{-1}$ in deionized water) by addition of a zinc chloride solution. pH treatments were achieved by amending the pH of the soil solution by additions of dilute hydrochloric acid or sodium hydroxide (in deionized water) to achieve deviations from ambient pH to +1, +2, +3, -1, -2, -3, and 0 (reference). For the temperature treatments, 10 ml of deionized water was added to jars which were incubated at a range of temperatures (20, 30, 37, 43, and 52°C) to achieve positive and negative deviations from ambient (37 °C). Moisture treatments were achieved by air drying soils for several days and adding various amounts of deionized water to approximately the dry soil to achieve 0.0, 0.05, 0.09, 0.17, 0.33, and 0.50 g water $g^{-1}$ soil (weight:weight) soil moisture treatments.

Six jars were prepared for each treatment for each of the two soils. All jars were purged with $N_2$ gas to remove oxygen, and three jars of each treatment received acetylene (10%). Jars were incubated for approximately 12 hr at ambient temperature (37°C; except for the temperature treatments) and the headspace was sampled several times to determine $N_2O$ production rates as described above. For the temperature treatments, jars were incubated for times ranging from 8 hr (43 and 52°C treatments) to 24 hr (20°C treatment) to allow for adequate biogeochemical activity and $N_2O$ production across the range of temperatures. The laboratory incubations for each of the five perturbation parameters were conducted on different days over a several week period and therefore the rates in the reference treatment for each perturbation assay should not be compared. Headspace samples were taken using 10 mL syringes with a gas-tight valve, and $N_2O$ was determined on an Agilent Technologies 6850 Series II electron capture gas chromatograph within a day of collection.

## 2.2 Estuarine Sediments – Salinity Pulse Perturbation

We examined the denitrifier perturbation response to pulse disturbance induced by a single physicochemical parameter (salinity) in environments that naturally experience a range in that parameter (estuarine sediments; Table 1). Sediments were sampled from three locations along the salinity gradient (ambient salinities of 0, 5, and 24 g $kg^{-1}$) in the Scheldt River estuary in Brussels and the Netherlands and we assessed rates of sediment denitrification and $N_2O$ production across a range of salinities (from 0 to 30 g $kg^{-1}$). Intact sediment cores were collected from freshwater (Appels, salinity 0 g $kg^{-1}$ at time of collection; 51.030309 N, 4.041905 E), oligohaline (Waarde, salinity 5 g $kg^{-1}$ at time of collection; 51.410664 N, 4.068669 E), and mesohaline (Rattekaai, salinity 24 g $kg^{-1}$ at time of collection; 51.449888 N, 4.195477 E) sites. The freshwater Appels site occupies the tidal freshwater region of the Scheldt River which is uniformly fresh (van Damme et al. 2005). The oligohaline Waarde site is in the Westerschelde Estuary into which the Scheldt River drains, with salinities ranging from 2 to 25 g $kg^{-1}$ (van Damme et al. 2005). The mesohaline Rattekaai site is located in the Oosterschelde, which Gerringa et al. (1998) report has higher salinities (around 30 g $kg^{-1}$) that are less variable because of little freshwater input.

Sediment cores were sectioned and approximately 2 g of surface (0-2 cm) sediment and 10 mL of water were placed into 38 $cm^3$ headspace vials. The salinity of the water added to the vials was amended by mixing 0.7 μm filtered freshwater (salinity = 0 g $kg^{-1}$; collected from the Appels site) and seawater (S = 30 g $kg^{-1}$; collected from the Scheldt Estuary). Sediments from

the freshwater Appels sites were incubated under salinities of 0 (ambient), 1, 3, 5, 10, 15, and 30 g kg$^{-1}$, the oligohaline Waarde site was incubated under salinities of 0, 1, 3, 5 (ambient), 10, 15, and 30 g kg$^{-1}$, and the mesohaline Rattekaai site was incubated under salinities of 0, 3, 5, 10, 24 (ambient), and 30 g kg$^{-1}$ (n=6 for each sediment/salinity treatment). All treatments also received 4 mM glucose and 2 mM $NO_3^-$.

After purging each vial with He to remove oxygen, the headspace of three vials for each treatment was amended with 10% acetylene. Vials were then incubated for approximately 24 hours at room temperature (20°C). Gas samples from the headspace of each vial were removed at several timepoints during the incubation into 10 mL evacuated headspace vials. The concentration of $N_2O$ was determined by electron capture gas chromatography on a Shimadzu GC8 gas chromatograph within one month of collection.

## 2.3 Estuarine Soils – Press-Pulse Salinity Perturbation

We investigated the response to long-term (~months) changes in physicochemical conditions (press perturbation) to contrast to the pulsed perturbation response described above (Table 1). Surface (0-2 cm) soils from a tidal freshwater marsh on the Delaware River estuary (Rancocas Creek; 39.9888002 N, 74.84483 W) were collected and 0.75 L of soil was mixed with 0.75 L of either artificial freshwater or saline water (using salts as described in section 2.1) to achieve long-term incubation (press) salinities of S = 0 g kg$^{-1}$ (control) or S = 20 g kg$^{-1}$ (press treatment). Duplicates of each treatment were incubated for 6 months in stoppered flasks with an oxygen-free headspace (purged with $N_2$ gas) under gentle mixing. To alleviate substrate limitation over the incubation period, the jars were amended by 0.4 mM $NO_3^-$ and 0.8 mM glucose weekly.

On days 0, 7, 14, 21, 35, 49, 70, 110, and 181, the long-term incubations were sub-sampled into smaller vials for short-term assays of denitrification and $N_2O$ production. 10 ml of the soil solution was sub-sampled into a 410 mL headspace vial, 10 ml of the appropriate salinity water was added to each vial along with 0.4 mM $NO_3^-$ and 0.8 mM glucose. Due to differences in starting salinity from the press treatments, the salinities after amendment that were assayed for the S = 0 and S = 20 press treatment were 0.0, 4.3, 7.6, 16.9, and 25.6 g kg$^{-1}$ (for S = 0) and 3.5, 7.4, 11.6, 20.0, and 28.3 g kg$^{-1}$ (for S = 20). Four oxygen-free vials for each press/pulse combination were prepared by purging the headspace with $N_2$, and acetylene (10% final volume) was added to 2 vials. Vials were incubated for <12 hr and the production of $N_2O$ was determined by subsampling the vials using 10 mL syringes with a gas-tight valve which were analyzed by gas chromatography (Agilent Technologies 6850 Series II electron capture gas chromatograph) within a day of collection.

On days 7, 35, and 110, immediately following the final headspace sampling for $N_2O$, the S = 0 and S = 20 press treatment soils that represented no pulse (salinity of 0 g kg$^{-1}$ for S = 0 and salinity of 20.0 g kg$^{-1}$ for S = 20) and pulse (salinity of 25.6 g kg$^{-1}$ for S = 0 and salinity of 3.5 g kg$^{-1}$ for S = 20) conditions were frozen at -80°C (samples without acetylene addition only). Nitrite reductase (*nirS*) and standard nitrous oxide reductase (*nosZ*) gene abundance (DNA), transcription products (cDNA), and expression (cDNA:DNA ratios) were measured on these soil samples. From each sample, DNA was extracted using the MoBio PowerSoil DNA isolation kit and RNA was extracted following a modification of the extraction methods described by Mettel et al. (2010) and Kearns et al. (2016) which uses Q Sepharose chromatography and is optimized for soils with high humic acid content. After extraction, the RNA was reverse transcribed to cDNA using Invitrogen's SuperScript III reverse transcriptase, following manufacturer's instructions. The concentration of DNA and cDNA was measured using Quant-iT PicoGreen and RiboGreen, respectively, following manufacturer's instructions and nucleic acids were normalized to 3 ng µL$^{-1}$, prior to amplification via quantitative PCR (qPCR) on a Stratagene MX-3005p quantitative thermocycler using *nirS* primers from Braker et al. (1998) and *nosZ* primers from Henry et al. (2006), following previously described protocols (Bowen et al.,

2011; Kearns et al., 2015). Standards for both genes were derived from purified PCR products and standard curves had slopes > 0.99, and amplification efficiencies of ~85%.

## 3 Results

Pulsed physicochemical perturbation elicited a short-term (~hours) response that resulted in reduced rates of denitrification with increasing levels of perturbation across the five mechanisms of perturbation investigated here. Both conventionally
farmed and organically farmed agricultural soils subjected to gradients of temperature, pH, toxicity (zinc), ionic strength (salinity), and moisture demonstrated reductions in rates of denitrification as the physicochemical variable deviated further from ambient conditions (Fig. 1). Rates of $N_2O$ production, in contrast, generally increased with increasing levels of perturbation (with the exception of zinc in the conventionally farmed soils; Fig. 1b), though $N_2O$ production rates exhibited a parabolic relationship with salinity (both soil types; Fig. 1a,f), zinc (for organically farmed soils; Fig. 1g), and soil moisture
(both soils; Fig. 1e,j) such that, at the highest levels of salinity and zinc and the lowest soil moisture treatments, we observed declines in net $N_2O$ production. There were no observed declines in $N_2O$ production with increasing deviation away from ambient temperature (for conventionally farmed soils; Fig. 1d) or pH (for both soil types; Fig. 1c,h). $N_2O$ production declined with increasing zinc in conventionally farmed soils (Fig. 1b) and declined with lower temperatures in organically farmed soils (Fig. 1i). In all cases, the $N_2O$:DNF ratio increased with increasing physicochemical perturbation, with $N_2O$ accounting for
between less than 10% (temperature) and nearly 100% (soil moisture and salinity) of total denitrification at the highest level of disturbance (Fig. 1).

In the experiment in which a salinity perturbation was imposed on estuarine sediments from three sites that had varying ambient salinities (0, 5, and 24 g kg$^{-1}$), we found the highest rates of denitrification at the ambient salinity with declining denitrification with deviation in salinity (Fig. 2). The lowest $N_2O$ production rates and $N_2O$:DNF ratios were likewise observed at ambient
salinities (Fig. 2). $N_2O$ production rates and $N_2O$:DNF ratios increased with deviation in salinity away from ambient conditions. The highest $N_2O$ production rate and $N_2O$:DNF ratios for freshwater sediments were observed at the highest salinity. In contrast, the highest $N_2O$ production and $N_2O$:DNF ratio was found in the lowest salinity for the mesohaline sediment (Fig. 2).

We investigated the response to long-term (~months) changes in physicochemical conditions (press perturbation) to contrast
to the pulsed perturbation responses described above. Soils from a tidal freshwater marsh (0 g kg$^{-1}$ ambient salinity) in the Delaware River estuary were incubated under anaerobic conditions for 6 months after adjusting the salinity to 20 g kg$^{-1}$ (the press treatment), along with a set of freshwater controls. Soils from these long-term incubations were subsampled throughout the 6-month period and assayed for rates of denitrification and $N_2O$ production when exposed to a range of salinities (approximately 0, 5, 10, 15, and 25 g kg$^{-1}$). The pulse perturbation response to this range of salinities in both the freshwater
control and saltwater-amended treatments was similar immediately following the initiation of the experiment (on day 0) and remained consistent in the freshwater controls throughout the 6-month experiment (as indicated by $N_2O$:DNF ratios; Fig. 3, black lines). In contrast, the response in soils subjected to the long-term press disturbance (the salinity-amended treatment) changed markedly over the 6-month period (Fig. 3, red lines). After 7 days at a salinity of 20 g kg$^{-1}$, the press treatment soils did not respond to changes in salinity ($N_2O$:DNF ratios were similar across pulse salinity treatments), though the consistently
elevated $N_2O$:DNF ratios (compared to the controls at a salinity of 0 g kg$^{-1}$) indicated a perturbation response across all pulsed

salinity levels (Fig. 3). At one month, the microbial community had adjusted to the higher salinity in the press treatment and exhibited a pulse perturbation response at lower salinities. This pattern was maintained for at least 6 months (Fig. 3).

The denitrifier gene expression in the press-pulse experiment demonstrated that *nirS* expression was not correlated with either $N_2O$ production or the $N_2O$:DNF ratio in either the controls or press treatment ($p > 0.05$; Fig. 4). In contrast, standard *nosZ*
expression was negatively correlated with the $N_2O$:DNF ratio in soils subjected to the press treatment ($p = 0.026$; Fig. 4). The relationship between *nosZ* expression and the $N_2O$:DNF ratio in control soils was similar, though the relationship was not significant ($p = 0.10$; Fig. 4). There was very little *nosZ* expression in any of the soils that experienced a pulsed change in salinity, either an increase in salinity for the controls or a decrease in salinity for the press treatment (Fig. 4).

**4. Discussion**

We found that changes in ionic strength (salinity), metal toxicity (zinc concentration), pH, temperature, and soil moisture all resulted in declines in denitrification, increased rates of $N_2O$ production (with decreased $N_2O$ production at higher levels of perturbation in some instances), and increased $N_2O$:DNF ratios (Fig. 1). There was a relatively consistent short-term response in rates of denitrification and $N_2O$ production in response to a wide range of physicochemical perturbations (Fig. 1). We propose that changes in physicochemical conditions can induce a generalized short-term perturbation response from the soil
denitrifying community, with higher $N_2O$:DNF ratios and increased net $N_2O$ production, with reductions in $N_2O$ production at higher levels of perturbation for some parameters (Fig. 5a). Physicochemical perturbation is defined here as a shift from the ambient physical and/or chemical conditions experienced by a soil microbial community.

There exists ample evidence that physical and chemical conditions influence denitrification and $N_2O$ production at the ecosystem scale. Temperature (Seitzinger, 1988; Larsen et al., 2011; Billings & Tiemann, 2014), salinity (Giblin et al., 2010;
Teixeira et al., 2013), pH (Firestone et al., 1980; Weslien et al., 2009; Baggs et al., 2010), toxic heavy metals (Magalhaes et al., 2007; Ruyters et al., 2010), organic pesticides (Yang et al., 2023), and soil moisture (Teh et al., 2011; Brown et al., 2012; Wang et al., 2023) have all been posited as important in controlling denitrification and $N_2O$ emissions in soils and/or sediments. Our findings, except for the press-pulse salinity experiment, are not applicable for elucidating the long-term controls of these environmental factors on denitrification and $N_2O$ production. Rather, this study expands our understanding of the short-term
response of the denitrifying community to alterations in the environment. Field measurements of $N_2O$ emissions have found pulses of $N_2O$ following physical disturbance of soils (Wang et al., 2005; Elder & Lal, 2008; Xu et al., 2015), soil thawing (Goodroad & Keeney, 1984; Chen et al., 2018), soil drying (Hou et al., 2012), clearcutting, and hurricane disturbance (Steudler et al., 1991). Our findings suggest a framework (Fig. 5) for a better understanding of the response of the denitrifying community to physicochemical perturbation.

While some of the physicochemical variables investigated here may have long-lasting effects on the denitrifying community and $N_2O$ production, such as low soil pH (Liu et al., 2010; Liu et al., 2014), there are others that might not exert impacts on denitrification and $N_2O$ production indefinitely. For instance, high or low salinity does not inherently induce a perturbation response. Rather, the deviation from *in situ* conditions creates a disturbance to which the microbial denitrifying community responds and recovers from (Fig. 5b), indicating that the perturbation response is relative to the background environmental
conditions experienced by the denitrifying community (Fig. 2). The press-pulse experiment further indicates that a microbial

community can become adjusted to a new physicochemical condition such that a return to the original condition, given enough time (about a month in this case), amounts to additional perturbation (Fig. 3).

The press-pulse experiment (Fig. 3) further indicates that initial perturbation confers subsequent resilience to further perturbation in the denitrifying microbial community (Fig. 5c; Philippot et al., 2008; Griffiths & Philippot, 2013). The
physicochemical pulse perturbation response consistently exceeds 60% $N_2O$ production at higher salinities in the freshwater controls (Fig. 3). In contrast, $N_2O$ production did not exceed 20% across all pulsed salinities after 6 months in the press treatments (Fig. 3). Similarly, the pulse perturbation response exceeded 50% $N_2O$ production in tidal freshwater sediments that do not normally experience fluctuations in salinity and remained below 20% $N_2O$ production in sediments from the oligohaline and mesohaline sites that experience daily (tidal) and seasonal fluctuations in salinity (Fig. 2). The observations from the
press/pulse experiment (Fig. 3) and measurements along an estuarine salinity gradient (Fig. 2) together suggest that denitrifying microbial communities that experience changing physicochemical conditions may be more resilient to subsequent disturbance than an undisturbed denitrifying community as would be found in more physicochemically stable environments (Fig. 5c). Further research to determine the generality of this finding across ecosystem types and forms of physicochemical perturbation is warranted.

An aspect that requires consideration is the methodological approach we used in the current study.  We utilized the acetylene block technique with the addition of substrates to soil or sediment slurries. The use of acetylene with the addition of substrates provides a measure of 'potential' denitrification rather than *in situ* rates of denitrification or $N_2O$ production (Groffman et al., 1999; Groffman et al., 2006). Acetylene is toxic to microbial nitrifiers (Hynes & Knowles, 1978) and potentially other members of the microbial community, which can inhibit coupled nitrification-denitrification and introduce other discrepancies that can
alter nitrogen cycling (Groffman et al., 2006). The use of soil/sediment slurries further alters the biogeochemical zonation found in soils and sediments that is critical to creating the conditions in which redox-sensitive nitrogen cycling processes proceed (Froelich et al., 1979). However, the acetylene block method remains a powerful tool for evaluating controls on denitrification (Groffman et al., 2009), and our approach allowed us to feasibly explore a range of physicochemical variables at various levels and sites (i.e., Figs. 1 and 2) and over time (i.e., Fig. 3) in a controlled setting that would be difficult to
undertake with other, less intrusive methods and with intact soils/sediments. Nevertheless, the generalizability of the perturbation response model to various physicochemical variables (Fig. 5) requires further investigation with less intrusive methodologies such as substrate isotope labelling (Nielsen 1992) that maintain microbial nitrogen cycling dynamics in relatively intact soils and sediments.

The physicochemical perturbation response we observed includes decreased rates of denitrification (Figs. 1 and 2), indicating
inhibition of some portion of the microbial community responsible for the reduction of nitrogen oxides to dinitrogen gas. The changing $N_2O$:DNF ratio, however, clearly indicates that the processes governing the production and consumption of $N_2O$ respond differently to the same physicochemical perturbation. Members of the microbial denitrifying community contain a large degree of modularity (Graf et al., 2014; Roco et al., 2017), possessing some or all of the genes that encode the enzymes necessary for catalyzing the nitrogen oxide reduction reactions. Changes in environmental conditions may promote modularity
or may drive shifts in these communities, both of which could result in an alteration of the $N_2O$:DNF ratio. For example, some denitrifiers lack the catalytic subunit gene for $N_2O$ reductase (*nosZ*) and produce $N_2O$ as the final metabolic product (Hedlund et al., 2011; Philippot et al., 2011). Complex microbial denitrifying communities in sediment/soil environments have been shown to include a variety of regulatory phenotypes that can result in the sequential transcription of genes in the denitrification pipeline, with transcription likely triggered by the production of intermediates (Liu et al. 2019). Further, an atypical *nosZ* gene
was identified that encodes a functional $N_2O$ reductase which, in many cases, is found in otherwise non-denitrifying organisms (Sanford et al., 2012), and the $N_2O$ uptake kinetics appear to differ between microbes with the standard and atypical *nosZ*

genes (Yoon et al., 2016). Fungal denitrification, in which $N_2O$ is the terminal product, has likewise received increased attention following the finding that some fungi are able to denitrify (Maeda et al., 2015). Rates of denitrification and $N_2O$ emission from soils have been linked to the structure of the denitrifying community (Cavigelli & Robertson, 2001; Ruyters et al., 2010; Philippot et al., 2011), and deviations in physicochemical conditions that promote or inhibit modularity, influence transcription, and/or select for certain portions of the denitrifying community may alter rates of denitrification and $N_2O$ emissions.

Community composition and relative gene abundance (Ruyters et al., 2010; Billings & Tiemann, 2014), transcription of the genes coding enzymes for $N_2O$ production and reduction (Magalhaes et al., 2011), and post-transcriptional interference with enzyme assembly and/or function (Liu et al., 2010; Liu et al., 2014) may all play a role in the observed $N_2O$ perturbation response. Our press-pulse experiment results indicate that the $N_2O$ salinity perturbation response is driven, at least in part, by inhibition of the expression of the nitrous oxide reductase enzyme, thereby resulting in a higher proportion of $N_2O$ as the final product (Fig. 5b). We observed low *nosZ* expression in all pulse treatments, and reduced expression in the press treatment that increased over time through the experiment (Fig. 4). We observed no correlation between *nirS* expression and either $N_2O$ production or the $N_2O$:DNF ratio (Fig. 4) across either press or pulse treatments. Further, *nosZ* expression was significantly correlated with the $N_2O$:DNF ratio in the press treatment (Fig. 4), suggesting that inhibited expression of the *nosZ* enzyme responsible for the reduction of $N_2O$ to $N_2$ was the likely mechanism for increased net $N_2O$ production in our press-pulse salinity experiment. In contrast, (Liu et al., 2014) found that post-transcriptional interference of *nosZ* enzyme assembly in low pH soils was the likely mechanism driving increased $N_2O$ production from soils. The mechanisms resulting in the $N_2O$ perturbation response may therefore differ between physicochemical variables, with likely combinations of both transcription and post-transcription enzyme inhibition together with more generalized impacts on the microbial community resulting in alternations to nitrogen cycling processes.

The pulse-press salinity experiment hints at the time required for the denitrifying community in estuarine sediments to adapt to salinity perturbation. We observed little *nosZ* expression in any of the pulse treatments with salinity perturbation whether from the control or press treatments (Fig. 4). In contrast, the *nosZ* expression in the press treatment without additional salinity perturbation recovered somewhat after one week at higher salinity (Fig. 4). Expression of *nosZ* appeared to have fully recovered by one month, with no further change observed (Fig. 4). Likewise, $N_2O$:DNF ratios in the press treatment demonstrated little response to pulsed changes in salinity after one week - the fraction of $N_2O$ produced is elevated (~20%) relative to the control without any pulse perturbation (~5%), but considerably lower than the control treatment at higher levels of salinity perturbation (which approach 100% $N_2O$; Fig. 3). At one-month post-press, the denitrifying community had adapted to the higher salinity and exhibited a perturbation response instead to reduced salinity, and the response was muted in comparison to the perturbation response of freshwater soils throughout the experiment (Fig. 3). There was little change in the salinity perturbation response in the press treatment after one month (Fig. 3). Both the $N_2O$:DNF ratios (Fig. 3) and *nosZ* expression results (Fig. 4) indicate that the microbial community recovered from the initial perturbation and adapted to the increased salinity level in the press treatment within one month, which suggests a generalizable model of perturbation recovery as gene expression changes with longer term changes in the microbial community (Fig. 5b). Chen et al. (2018) found that pulsed $N_2O$ emissions following thawing of soils extended for 18 days, which suggests a similar timeframe for a perturbation response to thawing. In contrast, Steudler et al. (1991) found higher $N_2O$ emissions for 7 months following a hurricane in subtropical forest soils. The transient perturbation response of the estuarine denitrifying community to salinity may be similar for some physicochemical variables, but it is unlikely to be the response to all changes in the environment. For instance, higher $N_2O$ production response was observed in soils that had been subjected to low pH conditions for over 20 years and was linked

to post-transcription inhibition of *nosZ* (Liu et al., 2010; Liu et al., 2014), indicating that the microbial community does not recover from all perturbations and the timing of any recovery might vary substantially.

## 5. Conclusions

Our research indicates that deviations away from physicochemical conditions to which the microbial denitrifying community is adapted can induce a perturbation response that promotes increased net $N_2O$ production over a broad range of environmental variables (Fig. 1). We suggest a generalized conceptual model of the physicochemical perturbation response characterized by declining denitrification accompanied by increases in the $N_2O$:DNF ratio, with increased net $N_2O$ production at moderate levels of disturbance (Fig. 5a). We show that the microbial denitrifying community may adapt to some physicochemical

variables over time, such as salinity (Fig. 3), with moderation of the pulse perturbation response under press disturbance conditions (Fig. 5b). The pulse salinity perturbation response is characterized by an initial inhibition of *nosZ* enzyme expression (Fig. 4) that gives way to more effective $N_2O$ reduction likely driven by recovery of gene expression and/or change in the denitrifier community composition (Fig. 5c). These findings indicate that an experimental press perturbation (Fig. 3) and *in situ* exposure to changes in physicochemical conditions such as salinity changes in oligohaline and mesohaline sediment

(Fig. 2) confer resilience to subsequent perturbation (Fig. 5b; Philippot et al., 2008; Li et al., 2014). We therefore hypothesize that the perturbation response will be stronger in denitrifying communities from physicochemically stable ecosystems (i.e., ocean sediments and deep tropical soils) than from ecosystems that experience more physicochemical variability (i.e., temperate soils and tidal marshes). It is likely that this generalized perturbation response model (Fig. 5) does not describe the response of the denitrifying community to all changes in the physicochemical environment (such as low soil pH; Liu et al.,

2014). However, this conceptual model may provide a useful framework for understanding (and potentially mitigating) $N_2O$ emissions from sediments and soils. Changing environmental conditions that perturb the denitrifying community likely promote 'hotspots' and 'hot moments' (Groffman et al., 2009) of $N_2O$ emissions and account for some of the variability in observed $N_2O$ emissions from soils and sediments.

## Appendix A: Example of nitrous oxide production

*Data Availability*. The data that support the findings of this paper are available in the Environmental Data Initiative repository at https://doi.org/10.6073/pasta/49650b321b5c977b5e8baa92d991254b (Weston 2024).

*Author contributions*. NBW, JLB, SBJ: Conceptualization. NBW, CT, PJK: Formal analysis. NBW, CT, PJK, WP, CH, CM, PVC: Investigation. NBW, JLB, SBJ: Methodology. NBW, SBJ: Project administration. NBW, CT: Visualization. NBW, CT, PJK, JLB, WP, SBJ: Writing – original draft. All authors: Writing – review and editing. NBW, JLB, PVC, SBJ: Funding
acquisition.

*Competing Interests*. The authors declare that they have no conflict of interest.

*Acknowledgements*. We thank M Garcia, T Hoffman, H Koch, A Laverman, JH Leaman, J Meschter, J Middelburg, G Ondrey, MA Vile, and J Walsh for assisting with field collection and laboratory analyses. This work was supported by the Department of Geography and the Environment at Villanova University (to NBW and CT) and the National Oceanic and Atmospheric
Administration Center for Sponsored Coastal Ocean Research/Coastal Ocean Program, through the South Carolina Sea Grant

Consortium, pursuant to National Oceanic and Atmospheric Administration Award NA960PO113 (LU-CES) and by the National Science Foundation (OCE-9982133 and OCE-0620959 to SBJ and DEB-1350491 to JLB) and the Georgia Sea Grant Program (awards NA06RG0029-R/WQ11 and R/WQ12A to SBJ).

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

**Table 1**. Summary of the three experiments in which the response of denitrification and nitrous oxide production to physicochemical perturbation was investigated (experiment number corresponds to the subsections in the Methods section).

| Expt. | Brief Description | Type of Soil/Sediment | Perturbation |
|---|---|---|---|
| 1 | Short-term (pulse) effect of perturbation by various physicochemical parameters | Agricultural soils (Lancaster, Pennsylvania, USA) | Salinity, pH, Zinc, Temperature, and Moisture |
| 2 | Short-term (pulse) effect of perturbation on sediments that experience natural variation in the perturbation parameter | Estuarine sediments (Scheldt River, Netherlands/Belgium) | Salinity |
| 3 | Long-term (press) effect and subsequent short-term (pulse) response to perturbation together with gene abundance and expression | Estuarine sediments (Delaware River, New Jersey, USA) | Salinity |


**Figure Captions**

Figure 1. Average (± standard deviation; n = 3) rates of total denitrification (DNF), nitrous oxide production ($N_2O$), and the $N_2O$:DNF ratios in conventionally farmed (a – e) and organically farmed (f – j) agricultural soils as a function of changes in salinity (a and f), zinc concentration (b and g), pH (c and h), temperature (d and i), and soil moisture (e and j). Quadratic equations have been fit through all data and significant relationships are indicated. The arrows denote increasing deviation away from *in situ* conditions (i.e., 'perturbation').

Figure 2. Average (± standard deviation; n = 3) rates of total denitrification (DNF), nitrous oxide production ($N_2O$), and the $N_2O$:DNF ratios in estuarine soils from tidal freshwater (*in situ* salinity = 0 g $kg^{-1}$), oligohaline (5 g $kg^{-1}$), and mesohaline (24 g $kg^{-1}$) sites in the Scheldt Estuary as a function of changes in salinity. The arrows denote deviation away from *in situ* salinity.

Figure 3. Average $N_2O$:DNF ratios (± standard deviation; n = 2) in a long-term (6 month) laboratory experiment in which tidal freshwater marsh soils were incubated under freshwater control (S = 0) or salinity-amended press conditions (S = 20). The soils were assayed for their short-term (pulse) salinity perturbation response on days 0 (immediately following salinity amendment), 7, 35, and 181 (results from sampling on days 14, 21, 49, 70, and 110 are included in Table S3). The horizontal dashed lines in the day 181 panel indicate the maximum $N_2O$:DNF ratios observed in the two treatments.

Figure 4. The relationship between $N_2O$:DNF ratios and *nosZ* enzyme expression from tidal freshwater marsh soils incubated under long-term freshwater control (S = 0) or saline press (S = 20) conditions and assayed on days 7, 35, and 110 with pulse or no-pulse salinity conditions. The timing of sampling (in days) is noted.

Figure 5. Conceptual model based on the results of this study that shows (a) relative rates of total denitrification (DNF), nitrous oxide ($N_2O$) production, and the $N_2O$:DNF ratio in sediments and soils as a function of a physicochemical perturbation gradient, (b) response of the denitrifying microbial community to physicochemical perturbation over time, and (c) the hypothesized relationship between ecosystem physicochemical variability and the perturbation response.

Figure A1. Example of timecourse concentrations of $N_2O$ in the headspace of soil incubations for the determination of $N_2O$ production and denitrification (DNF; $N_2O$ + $N_2$ production measured by amendment with acetylene). These soils are from the temperature assay with agricultural soils.

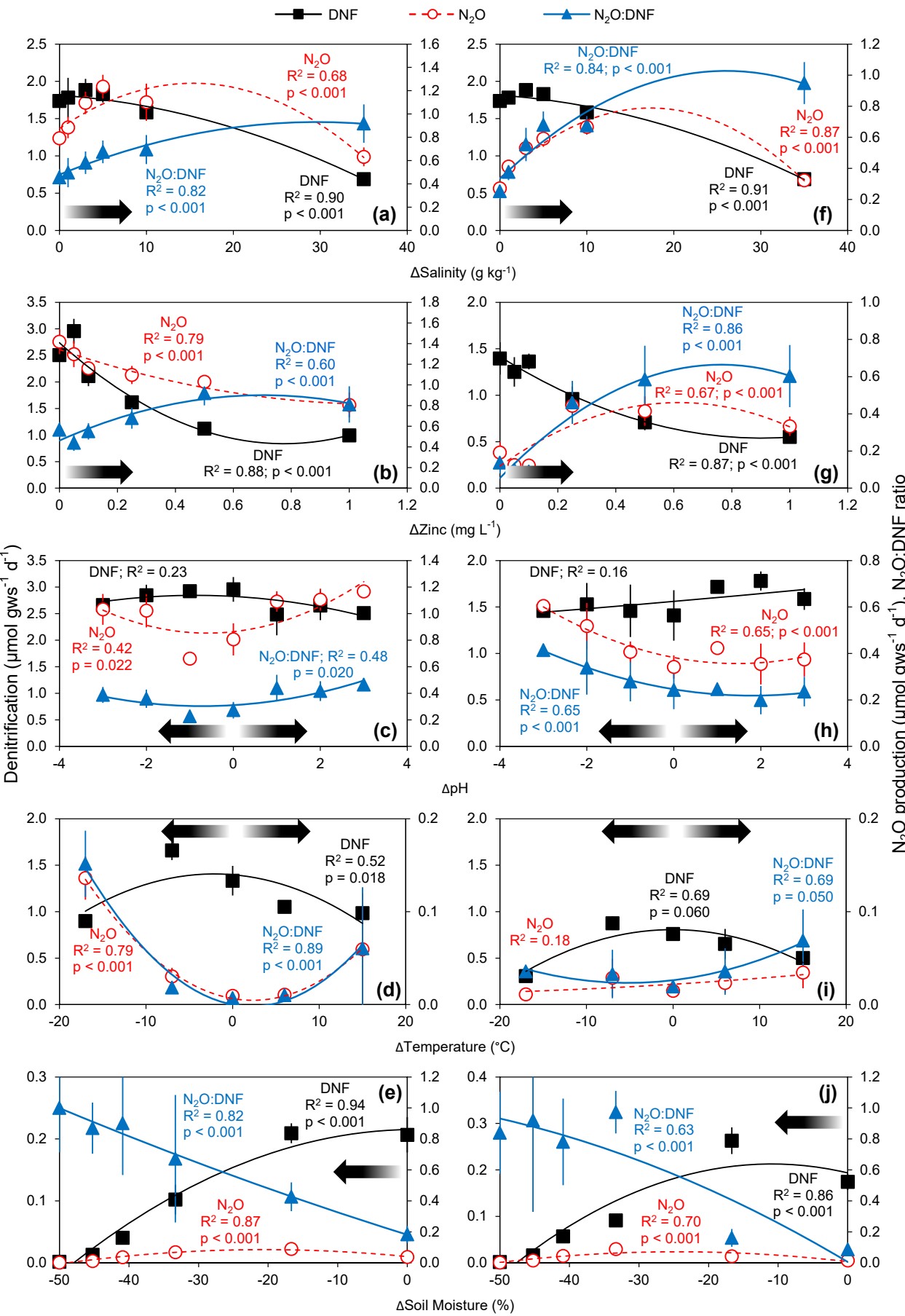

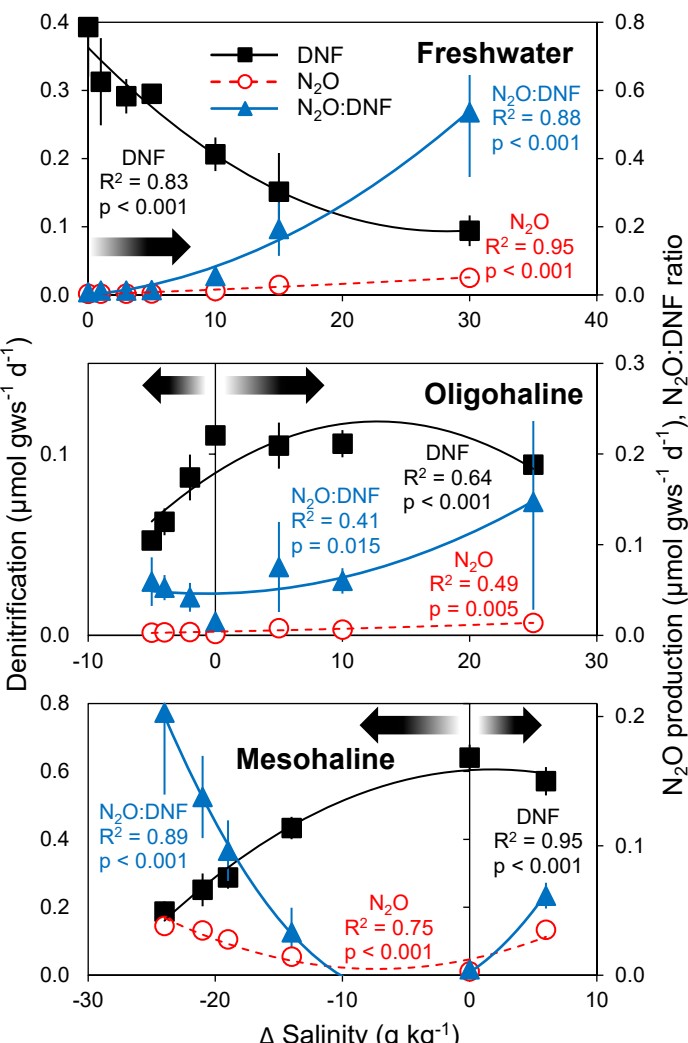

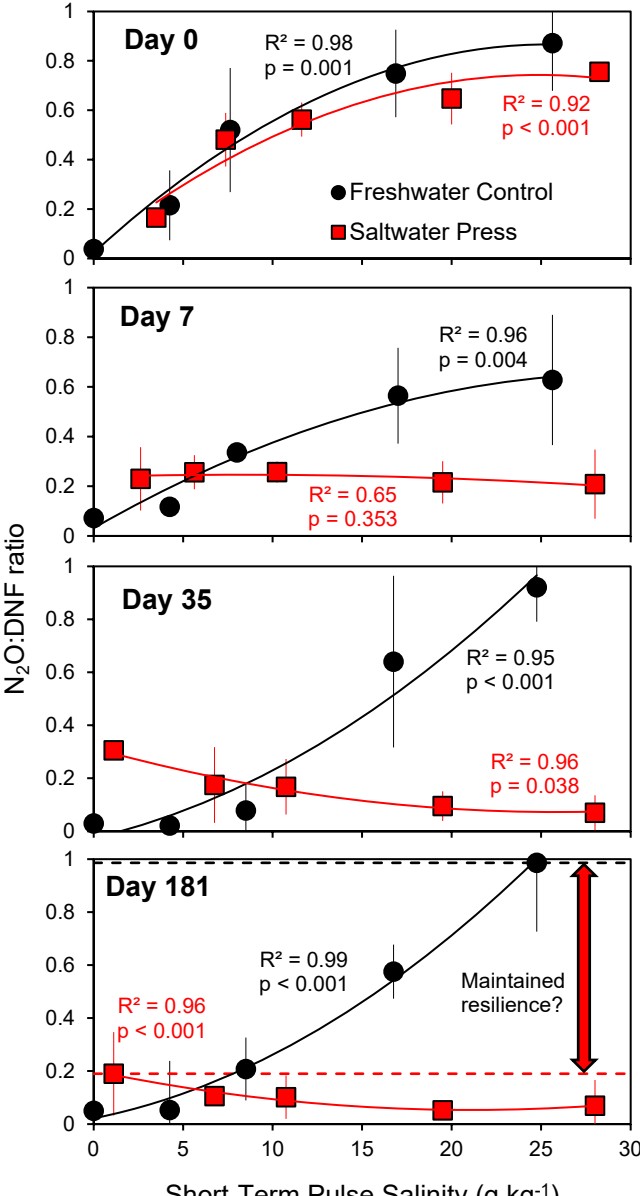

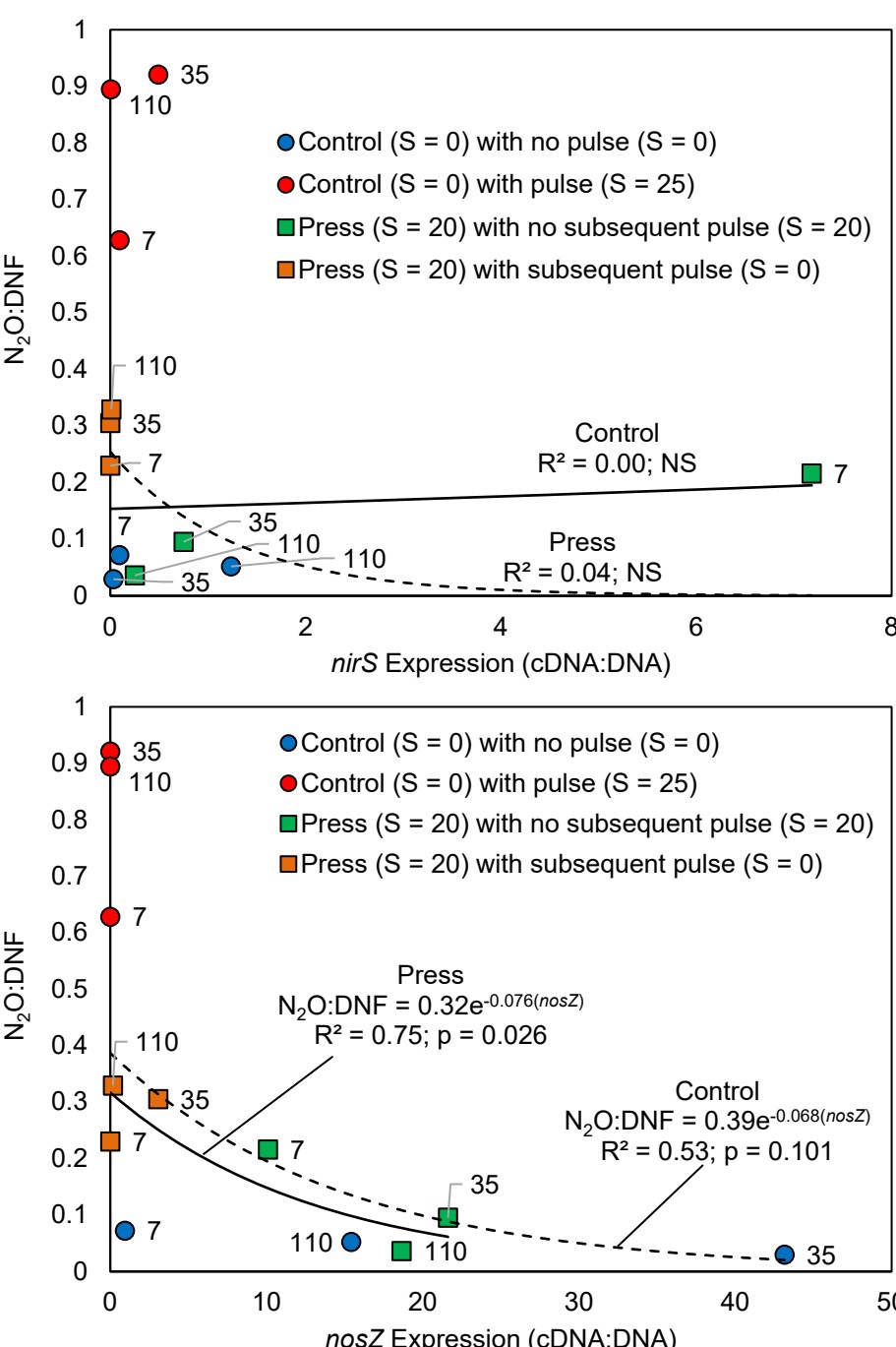

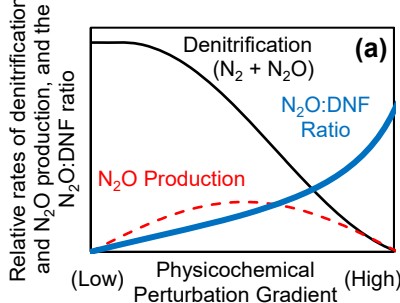
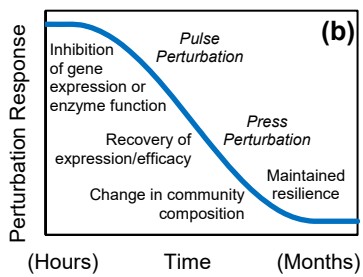
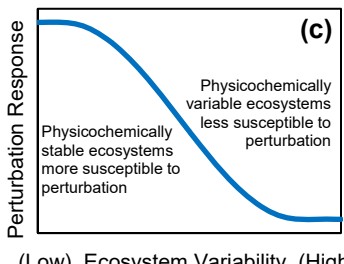

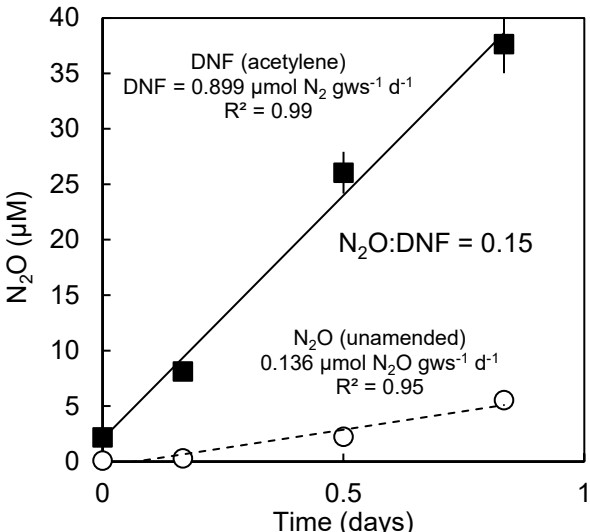

Figure A1