# Peer review of "Physicochemical Perturbation Increases Nitrous Oxide Production from Denitrification in Soils and Sediments"

_EGUsphere, 2024_

## Author Comment (AC1)

The authors thank Dr. Thibault de Chanvalon for his thoughtful comments on our manuscript. The reviewer's comments and suggested revisions will improve the final version of the manuscript. Our responses are in blue, below.

The manuscript investigates the change of N2O production and denitrification (DNF) in slurries after the physicochemical perturbations artificially produced. In general, the different kind of perturbations induce an increase of the N2O/DNF ratio during about one month. Based on estuarine slurries the authors also demonstrate that communities adapted to changing environment are more adapted to the changes.

The approach seems original and provides important information about the variability of N2O emissions. An important effort is made to keep the result and discussion synthetic in order to focus the reader's attention on the concepts presented, well summarized in the figure 5.

However, before publication, it seems important to lift three concerns I have concerning the experimental design:

1 – All the rates measured assume linearity of the N2O increase. From my experience, it is a transient species i.e. after a burst of NO3 it is going to increase then decrease. You wrote it increases linearly but did you measure it ? (it seems so for the pulse experiment but no time series is shown in the result) If not how can you justify your assumption?

We measured $N_2O$ increase over time in these various experiments, typically measuring four to five timepoints during each incubation. We report the final rates calculated from the slope of the increase in $N_2O$ over time in our manuscript. An example of our timecourse data is shown below (Fig. RC2a). We will clarify that we calculate rates using timecourse data and include this figure in the supplementary information in the revision.

[Figure]

Figure RC2a. Example of timecourse concentrations of $N_2O$ in the headspace of soil incubations for the determination of $N_2O$ production and denitrification (DNF; amended with acetylene). These soils are from the temperature assay with agricultural soils (experiment 2).

For the increase to be linear, it is necessary to measure the initial rate, at least when less than a third of the NO3- has been consumed, however when I try to calculate mass budget of N in your experiment, it appears that almost the entire stock of nitrate have been consumed (for the higher rates reported). Can you give a maximum rate measurable for each of your experimental conditions? Did you measure the NO3 changes? Why do you not consider the importance of possible other intermediate species contributing to DNF such as NO2-?

Exp1

20g x 1 umol/g/d x 12h= 10umol

10mL x 1mmol/L = 10 umol

Exp 3

10mL (g?) of slurry diluted by 2 = ~5g x 2 umol/g/d x 12h = 5 umol

10mL x 0,4 mM = 4 umol

For experiment 1, the author is correct that there was 10 umol $NO_3^-$ available in the slurry and that a rate of 1 umol gws-1 d-1 would consume that amount of $NO_3^-$. For experiment 3, there was 8 umol $NO_3^-$ available (10 ml slurry + 10 ml amendment was brought to 0.4 mM) and a rate of about 1.5 umol gws$^{-1}$ d$^{-1}$ would have consumed the addition during the full incubation period. As noted above, however, we measured $N_2O$ and DNF over time during the incubation period, typically at 0, 2, 4, 8, and 12 hours. If

rates were high enough that substrate limitation was an issue and the production of $N_2O$ levelled off over time, we removed those timepoints from the rate calculation. The maximum rates measurable are therefore higher than any rates we report here. For instance, in experiment 1, rates would need to exceed 6 umol gws$^{-1}$ d$^{-1}$ to result in substrate limitation prior to determination of the linear slope of increase in $N_2O$. In experiment 2, the rates would need to be greater than 8 umol gws$^{-1}$ d$^{-1}$ to exceed our ability to calculate the linear slope of increase. Rates were consistently lower than these values. We are confident that substrate limitation did not influence the rates we report here.

It is currently not clear in the manuscript that we measured $N_2O$ over time during the incubation periods and used the linear increase to determine the rate, which led to the concern about substrate limitation and nonlinearity in $N_2O$ production the reviewer outlined above.  We will include more specific information about the timecourse sampling for each experiment in the manuscript revision. We did not measure the change in $NO_3^-$ or $NO_2^-$ concentration in our experiments. The measurement of $NO_3^-$ or $NO_2^-$ may have supplied additional interesting information about $NO_3^-$ uptake and the concentration of the intermediate ($NO_2^-$) but would not, ultimately, inform our conclusions about $N_2O$ production and denitrification.

2 – The main goal is to understand the effect of an artificial perturbation on N2O production. To this aims the authors compare N2O production of perturbed slurry to a reference. However, the reference itself seems strongly perturbated by the experimental design. In particular, the anoxic conditions produced modify the community structures. Similarly, glucose addition would favor opportunistic species. Have you performed the experiment without these two modifications to obtain a more realistic reference rate measured? Why do you think it is mandatory to use these two perturbating conditions?

We agree that the experimental design that includes soil slurries, amendments of nitrate and organic matter, and the addition of acetylene may result in perturbation in addition to any perturbation caused by the changes in other physicochemical conditions. We include a paragraph in the discussion that outlines these concerns (lines 249-261). We fully agree that additional studies with methodology that retains undisturbed soil structure without amendments of nitrate or organic matter is required. It is possible that the perturbation response we outline here would be even higher since, as the reviewer notes, our control soils are perturbed by our design. Approaches using, for instance, isotopically labeled N to track denitrification and $N_2O$

production would be logical next steps to further evaluate the perturbation response we propose here.

We have explored the role of organic matter additions on the rates of denitrification and nitrous oxide production as outlined in this manuscript. For instance, we conducted a set of experiments with sediments from the Scheldt River without organic matter additions and with glucose, acetate, and lactate added at various concentrations (Fig. RC2b). We found that, while rates of denitrification increased with organic matter additions and responded to the type of organic matter (glucose>acetate>lactate), nitrous oxide production did not change appreciably and the $N_2O:DNF$ ratio decreased with the addition of organic matter (Fig. RC2b). We felt that this data, while interesting, is not directly related to the perturbation response hypothesis we are reporting in the manuscript and so does not belong in the revision. However, these findings would suggest that the perturbation response might be even stronger than we report here given that glucose amendments were made to all three of the experiments we report here.

[Figure]

Figure RC2b. Rates of denitrification (DNF; N2+N2O), N2O production, and the N2O:DNF ratio in estuarine sediments (Scheldt River) incubated without organic matter addition (Ambient), and with 1mM or 5mM additions of glucose, acetate, or lactate.

We likewise explored the response of nitrate availability on rates of denitrification and $N_2O$ production (Fig. RC2c). Higher $N_2O$ production in response to greater nitrate availability is a well-known phenomenon (e.g., Firestone et al., 1980), and we observed higher $N_2O$ production with increased nitrate availability even as rates of total denitrification reached their maximum at lower levels of nitrate availability (Fig. RC2c). We do not feel that this figure, which confirms a relatively well-known response, is needed in the revision.

[Figure]

Figure RC2c. Rates of denitrification (DNF; N2+N2O), N2O production, and the N2O:DNF ratio in estuarine sediments (Scheldt River) at various nitrate concentrations.

We elected to use concentrations of nitrate low (<2 mM) relative to the tens of mM that elicit high $N_2O$ production irrespective of perturbation effects in our experimental design.  The sediments we used for the perturbation experiments were reducing wetland sediments that has very low ambient nitrate, and the acetylene method (anoxic conditions and the presence of acetylene) does not allow for nitrification. The addition of nitrate is meant to alleviate substrate limitation and produce a "potential" rate of denitrification. As stated above and in the manuscript, building on this investigation of the role of perturbation on nitrous oxide dynamics with methods that do not require amendments or manipulation of soils/sediments is needed. We do feel, however, that the current manuscript outlines a relatively novel way of better understanding $N_2O$ production dynamics that, even with some methodological challenges, warrants publication.

3 – In the first experiment, the reference incubation (for ΔT= ΔZinc = Δmoisture = ΔSalinity = ΔpH =0) for the soil treatments present in Figure 1 should correspond to the same slurry. However, very different production rates are reported (from 0 to 0.8 for N2O production and from 0.2 to 2.8 for DNF) which cast serious doubt about the reproducibility of the experiments, and the validity of the experimental design.

The moisture treatment entailed air-drying soil prior to amending soil with water to achieve various soil moisture treatments. The drying of the soil prior to the rate measurements likely resulted in the lower rates measured in this treatment compared to the other treatments. The reference condition does not differ as much for the other treatments, though we agree there is some variation. These differences may stem from the timing of the incubations and the condition of the soil. Given the large number of

treatments (a total of 360 incubations were performed with more than 1400 $N_2O$ samples analyzed on the gas chromatograph in the first experiment), each of the five perturbation incubations were performed at different times. The incubations were all performed within a 2 month period, but changes in the environmental conditions of the source soil may have influenced the reference rates of DNF and $N_2O$ production. We do not make any comparisons of rates across treatments, but rather we compare rates of DNF and $N_2O$ production within each incubation to elucidate the role of perturbation relative to the reference. The differences between the rates in the reference conditions in each treatment, therefore, are not of concern. We will clarify this in the revision.

Additionally, the methods need much better description of the methodology used and the associated limitation (see below). It could be done in a supplementary file.

I would also appreciate the author to propose a hypothesis about the underline mechanism responsible for such common type of answer: does N2O producer adapt faster than N2 producer? Is it due to thermodynamic barrier?

We outline the underlying mechanism(s) that we argue are likely responsible for the observed perturbation response in the final paragraph of the discussion (lines 313-331). The nosZ and nirS expression results point to inhibition in the expression of nosZ and not of nirS following salinity perturbation. This is followed by a recovery in nosZ expression together with a reduction in additional perturbation response, indicating that initial perturbation confers resilience on the microbial community. Whether these responses are due to changes in enzyme expression within the existing microbial community or changes to the overall microbial community (or some combination of the two) is beyond the scope of our study.

Details remarks:

Sites characterization:

Why selected the 0-2 cm depth layer? did you check the absence of oxygen and the decrease of NO3 to identify the layer with the most active denitrifying community?

We did not verify absence of oxygen or nitrate update rates when selecting the soil/sediment depth, but surface soils are typically the most biogeochemically active layer within the soil/sediment column.

Do you have other ancillary parameters that could give some chemical context:

Do you know the natural NO3 concentration, does it vary between your samples, could that play a role?

Did you characterize by any way the natural heterotrophic activity? (organic matter lability? Oxygen consumption -DBO5 of your slurries? Enzymatic activities?)

Did you measure other elements that could interfere in the N cycle such as redox element (Fe, H2S, ...) ? or other N species that could better describe the natural N cycle of your site such as NO2- ?

We did not measure these soil/sediment characteristics uniformly across the sites/experiments. We argue that these parameters would be important in comparing rates of denitrification and/or $N_2O$ production across sites, they are less important in understanding the perturbation response in a laboratory setting as reported here.

Incubation Design:

For experiment 3: "the jars were amended to a final concentration of 0.4 mM NO 3- and 0.8 mM glucose weekly" it is not clear if the chemicals were added to increase the slurries concentration *by* 0.4 and 0.8 mM or *to* 0.4 and 0.8 mM (which implicate you measured the NO3 and glucose concentration)

The slurries were amended by 0.4 $NO_3^-$ and 0.8 mM glucose weekly. We did not measure the glucose levels.  This will be corrected in the revision.

For experiment 1 and 2, 10mL of water is mixed with 2g of soil or sediment, where does the water come from?

In experiment 1, the water mixed with soils was deionized water amended with nitrate and glucose. In experiment 2, the water was site water collected from the sites and diluted with water from the Apples site (freshwater) or seawater to achieve the target salinity. This will be detailed in the manuscript.

Is the use of N2 purge to take off O2 could change your community behaviour ? Why do not use Ar instead ? why did you use He for the second set of experiment?

He was used in the 2$^{nd}$ experiment versus $N_2$ in the 1$^{st}$ and 3$^{rd}$ experiments simply because of the various resources available to us in the laboratories in which we were conducting our research (Villanova University in Pennsylvania, USA for experiment 1 and 3, and the Netherlands Institute of Ecology and the University of Georgia, USA for experiment 2). $N_2$ is ~80% of the atmosphere that these soils/sediments are exposed

to, and we expect that purging oxygen with 100% $N_2$ would not change the microbial community behavior relative to argon or helium.

Analytical strategy :

How did you took your aliquots from your jars for N2O analyses? is there any risk of O2 contaminations during this sampling, in particular for pulse experiment where a time serie was performed with He in the headspace (a very volatile gas)? What the introduction of O2 could produce? Did you check the absence of oxygen ?

For experiments 1 and 3, headspace samples for $N_2O$ analysis were removed using 10 ml syringes (with valve) and injected into gas chromatograph within hours of collection. The system we use is gas tight, though there is always risk of oxygen contamination. The introduction of a small amount of oxygen is unlikely to alter results in a significant way. Denitrifiers are typically not as sensitive to oxygen as microbes performing terminal respiration with more reduced electron acceptors (such as sulfate reduction and methanogenesis), and oxygen would be quickly consumed in the incubation jars. We carefully reviewed the production of $N_2O$ in each incubation jar (i.e., Fig. RC2a) that would indicate nonlinearity in $N_2O$ production in response to any major disruption (in addition to substrate limitation). We do not feel that the introduction of oxygen in these experiments was likely. We will clarify the details of the sampling in the revised manuscript.

How long did you store your gas aliquots before measurement ? In which vessel ? what is the detection limit of your instrument ? are you always far above the detection limit?

The samples in experiment 1 and 3 were analyzed within hours of collection (they were sampled into 10 ml syringes with a valve). The samples in experiment 2 were sampled into gas-tight, evacuated vials with a septa closure. These samples were stored for several weeks prior to analysis. These details will be clarified in the revision. The electron capture detector gas chromatography approach has very low detection limits. While our initial samples during the timecourse measurements (i.e., Fig. RC2a) are sometimes below detection, the production of $N_2O$ was easily detectable in subsequent measurements.

How did you estimate the mass of sediment? Is it the mass of solid calculated from porosity or the mass of the bulk sediment (ie mainly water)?

We report values as rates per gram fresh/wet sediment/soil. This will be clarified in the text of the manuscript.

Writing:

In figure 2, I recommend to put two black lines in the oligohaline and mesohaline sites to delimitated the daily changes of salinity.

This is a good suggestion. However, we do not have data sufficient to bracket salinity at these sites on a daily basis. Salinity changes significantly over seasonal cycles as well due to changes in discharge. In the text, we will include additional information about the salinity range at these sites.  From van Damme et al. (2005; 10.1007/s10750-004-7102-2), we can say that the Appels site is uniformly fresh and salinity ranges from 2 to 25 psu at the Waarde site. There is less information available about the Rattekaai site in the Oosterschelde, but Gerringa et al. (1998) suggests the salinity is generally around 30 psu with less seasonal and daily variation.

For the title of sections 2.1 and 2.2, I suggest to replace "perturbation" by "Pulse disturbance" in order to keep the same expression along the entire manuscript.

This is a good suggestion. We will make this revision as recommended.

---

## Author Comment (AC2)

We thank the reviewer for their thoughtful comments on our manuscript. The reviewer's comments and suggested revisions will improve the final version of the manuscript. Our responses to reviewer comments are in blue.

This manuscript describes the results of several batch experiments performed with the objective of evaluating N2O emissions response to different environmental stressors. The tested stressors were: salinity, pH, temperature, moisture and Zn. The reported results include the measurement of accumulated N2O in the head-space of the reactors for the different tested treatments and the detection of nirS and nosZ genes expression. Also, the N2O to N2 ratio is estimated because acetylene is added in some reactors to stop N2O reduction. The methodological approach is well described and the obtained results and conclusions are clear. The findings allow an improved understanding on the role of environmental perturbations on N2O emissions.

Nevertheless, I would kindly ask the authors to address the following issues:

1. In the present study, three different approaches are employed to study the role of salinity changes on N2O emissions. Instead, a single approach is employed for the other tested parameters (pH, temperature, moisture and Zn). Which is the reason of using different methodologies for different parameters? Which has been the reason of choosing these parameters instead of others (e.g. pesticides)? Also, regarding Zn, why this metal instead of others? All the responses might be related to the study site but it is not clear enough in the manuscript.

   The initial experiment in which we tested multiple types of perturbation (pH, temperature, soil moisture, Zn, and salinity) was focused on understanding how the denitrifying community responds to various physicochemical perturbations. There are indeed additional parameters that could have been investigated such as pesticides or other toxic heavy metals, and it was simply a decision around keeping the experimental design feasible. Future studies might expand the tested parameters. The subsequent experiment investigated how denitrifying communities from environments experiencing a range in one of those parameters (salinity in estuarine sediments) respond to changes in that parameter, and the final experiment evaluated the long-term response of the denitrifying community to a change in that same parameter. We elected to focus on a single parameter in these sets of experiments for logistical reasons – we did not have the resources to investigate multiple parameters. We elected to focus on salinity for the additional experiments for several reasons, including 1) it is a parameter that changes over daily (tidal) and seasonal timescales in estuarine environments and is therefore ecologically relevant, 2) it is a parameter that will

be altered in some environments in response to climate change, 3) it is relatively easy to manipulate and to measure. We readily acknowledge these same arguments could be made in favor of other parameters such as temperature and pH, and that additional experiment using these (or other) parameters were beyond the scope of the current study. We will clarify these points in the opening paragraph of the methods section of the manuscript.

2. In the methodology section concerning the agricultural soils tests (2.1), you explain that you tested conventional and organic farming practices. However, nothing more is said about it on the results and discussion sections. Could you explain why? Also, to which type of soil correspond the results you are showing?

   The data shown in Figure 1 is for the conventionally farmed soil. It was an oversight that we did not include that information in the figure caption, and we will do so. The full data from both sets of experiments is included in the appendix, but we elected to show results from one soil type and did not address the differences between the conventional and organic soils because the response of both soil types to the various physicochemical perturbations is remarkably similar. We will make that point more clearly in the revised manuscript.

3. It is well stated that all experiments were performed under anoxic conditions. However, which is the gas replacing oxygen in the section 2.3 experiments? Why did you use N2 for the experiments described in 2.1 while He was used in the experiments described in 2.2? The same issue is found regarding the gas chromatography methodology employed for N2O measurements for each set of experiments. In 2.1, an Agilent GC is used, in 2.2 it is a Shimadzu GC, while no information is provided for 2.3 experiments.

   We used $N_2$ to create anoxic conditions in the section 2.3 experiment, and $N_2O$ was analyzed on an Agilent GC in section 2.3.  We will include that information in the revised manuscript. These differences are simply the result of working in different laboratories (Villanova University in Pennsylvania, USA for sections 2.1 and 2.3, and the Netherlands Institute of Ecology and the University of Georgia, USA for section 2.2) with different resources for different portions of the work described here, and we don't expect that the differences in headspace gas or gas chromatography influenced the results.

4. In several sentences along the manuscript (e.g. lines 83 and 160) the authors associate the nirS gene to a nitrate reductase. This is not correct since nirS gene

is associated to nitrite reductase (as well as nirK). In this context, why checking the nitrite reductase instead of the nitrate reductase or maybe both?

Thank you for catching this mistake. We define nir as nitrite reductase initially but then made the mistake of calling it nitrate reductase in several spots. We will correct this typo in the revised manuscript. We elected to focus on nir expression because it is downstream of the nar nitrate reductase enzyme and is the first step in the overall denitrification pipeline that produces a gaseous end-product (nitric oxide). The reduction of nitrate via the nar enzyme may lead to annamox, an alternate pathway of nitrite reduction. The measurement of nir nitrite reductase enzyme expression thus more fully isolates steps in the denitrification pathway.

In the following sentences, you will find some additional specific comment where numbers correspond to the manuscript lines:

1: It should maybe be mentioned in the title that the study focuses on nitrous oxide production from denitrification because other pathways of nitrous oxide production are not considered (e.g. nitrification or chemodenitrification).

This is good point of clarification. We will change the title to "Physicochemical Perturbation Increases Nitrous Oxide Production from Denitrification in Soils and Sediments"

46-48: Even I understand it's not the object of the study, it might be worth it to mention that N2O emissions can also be derived from the nitrification and chemodenitrification processes apart from denitrification. In fact, the contribution of chemodenitrification on N2O emissions is not still well known but could be a significant source. E.g. Robinson et al. (2021) https://doi.org/10.1039/D1EM00222H; Jones et al. (2015) https://doi.org/10.1021/es504862x; Cooper et al. (2003) https://doi.org/10.1128/AEM.69.6.3517-3525.2003 .

We mention abiotic denitrification in this section which is a synonym for chemodenitrificaiton, but will clarify that statement to include chemodenitrification.

51-52: How many decades exactly?

The rate of 0.26% per year increase in $N_2O$ concentrations is based on data from 1980-2005 (Forster et al. 2007). We will clarify this in the revised manuscript.

62-63: What about the role of inorganic electron donors such as ferrous iron or sulfide?

Yes, this is a good point – we will include a statement about the availability of electron donors.

68: Due to incomplete denitrification?

Possibly, but the mechanism is unclear. Burgin and Groffman (2012) suggest that the presence of $O_2$ selects for microbes that produce more $N_2O$. It may also be inhibition of nitrous oxide reductase (Chen et al. 2023; https://doi-org.ezp1.villanova.edu/10.1021/acs.est.2c07081) in a mechanism somewhat similar to the more generalized physicochemical perturbation response we posit in this manuscript.

71-72: Why?

Lower pH has been linked to the posttranscriptional inhibition of nitrous oxide reductase (Liu et al. 2010). This will be clarified in a statement in the manuscript.

74: You have already explained the potential role of soil moisture in the previous paragraph. Try to avoid repeated information.

This will be addressed in the revision.

102-171: Even the methodology description is very clear, a table summarizing all tested conditions could be included in the revised version of the manuscript to facilitate following the results and discussion sections.

We will include the following table in the revised manuscript:

Table 1. Summary of the three experiments in which the response of denitrification and nitrous oxide production to physicochemical perturbation was investigated (numbers corresponding to the subsections of the Methods section).

| Expt. | Brief Description | Type of Soil/Sediment | Perturbation |
|---|---|---|---|
| 2.1 | Short-term (pulse) effect of perturbation by various physicochemical parameters | Agricultural soils (Lancaster, Pennsylvania, USA) | Salinity, pH, Zinc, Temperature and Moisture |

| 2.2 | Short-term (pulse) effect of perturbation on sediments that experience various ranges in the perturbation parameter | Estuarine sediments (Scheldt River, Netherlands/Belgium) | Salinity |
| 2.3 | Long-term (press) effect and subsequent short-term (pulse) response together with gene abundance and expression | Estuarine sediments (Delaware River, New Jersey, USA) | Salinity |

99: Gram soil/sediment per day --> dry or fresh soil/sediment?

All rates are per gram fresh (wet) soil/sediment. This will be clarified in the revised manuscript.

108+125: Units for salinity are missing

This will be clarified in revision.

119-120: Jars were incubated 12h except for the temperature treatment. Why did this treatment have a different incubation time?

Cooler temperatures were incubated for longer periods of time (24 hr) to allow for adequate biogeochemical activity, while warmer temperatures were incubated for short periods of time (8 hr) because of higher biological activity. This will be clarified in the paper.

177: For Zn this is not observed for the first 3 points, right?

Yes, this is true. We will amend the sentence to clarify the pattern of $N_2O$ production with increasing zinc concentration.

188: Fig.3 --> Fig.2 ?

Correct. Thank you for catching that.

190: "above43" --> above

This will be corrected.

215: I think "moderate" is not enough specific here

We will reword this statement.

We propose that changes in physicochemical conditions can induce a generalized short-term perturbation response from the soil denitrifying community, with higher $N_2O$:DNF ratios and increased net $N_2O$ production with reductions in $N_2O$ production at higher levels of perturbation for some parameters (Fig. 5a).

231: "do not appear to" or "might"?

This will be amended.

248: Only across ecosystem types or also across stressor types?

Yes, good suggestion. We will amend this statement to "Further research to determine the generality of this finding across ecosystem types and forms of physicochemical perturbation is warranted."

249-261: Could an isotope labelling approach (15N) be more appropriate than the acetylene block method?

Yes, an isotope labelling approach would be a good way to further address the role of physicochemical perturbation. That is what we had in mind in writing this section, though we don't state it clearly. We will edit this paragraph to include the suggestion that isotope labeling methods would be an appropriate next step.

262-277: Concerning the succession of active denitrifying microorganisms, I think it is also worth having a look at the paper published by Liu et al., in 2019 (DOI: 10.3389/fmicb.2018.03208).

Agreed, this is a good study to reference in this section. We will amend this portion of the paper and include the citation to Liu et al. 2019.

284: No pattern or flat pattern?

We will clarify this statement. "We observed no correlation between nirS expression and either $N_2O$ production or the $N_2O$:DNF ratio."

292-229: In general, the information included in this paragraph seems somehow repetitive with respect to the statements made on the two previous paragraphs. Consider summarizing it.

We will attempt to shorten this paragraph in the revision. The purpose of this paragraph is to address the timeframe of the perturbation response, which is not addressed in the previous paragraphs. We will remove any redundancy with previous paragraphs.

319: "nos" --> nosZ ?

We will correct this.

Figure 1 to 4: why did you choose to fit quadratic equations? In some cases, it does not reflect the evolution of the measured parameters and induces confusion on the results interpretation.

We agree that not all relationships require nonlinear fits (linear fits would work in some instances), but many of the relationships are nonlinear and we felt it would induce

more confusion to use different types of fits than to be consistent. We argue that the current approach causes the least confusion.

Figure caption of Figure 1: I don't understand to what refers the "1" for standard deviation.

This was meant this to clarify that we meant 1 standard deviation (rather than 2 or 3), but we agree that it creates confusion rather than reduces it, and we will remove the number. All error bars are a single standard deviation.

Figure 4: Why did you prefer to draw joint instead of separate lines for press and control experiments? Could it be useful to include the nirS results in this Figure?

In a revised Figure 4 (below), we have shown separate lines for the press and control experiments in the nosZ to $N_2O$:DNF figure, and we have included the nirS to $N_2O$:DNF figure as well showing no relationship. The separate lines in the nosZ portion of the figure show a significant relationship for the press data and a very similar but non-significant relationship for the control data. The updated figure will be included in the revision, and we will amend the text in the manuscript as appropriate to cite this figure.

[Figure]

Figure 5: I think you should state in the figure caption that this model has been performed according to your results. Maybe you could also check if it fits what has been found by other authors to perform a scheme according to all literature available up to date (including your study)? In plots b and c, following your conclusions maybe the slope at the right side should be softened to clarify adaptation after time.

We have softened the slope in panels b and c in Fig. 5 to clarify maintained resilience in the $N_2O$ perturbation response as suggested. The new figure, copied below, will be included in the revision. We will change the figure caption as follows:

Figure 5. Conceptual model based on the results of this study that shows (a) relative rates of total denitrification (DNF), nitrous oxide ($N_2O$) production, and the $N_2O$:DNF ratio in sediments and soils as a function of a physicochemical perturbation gradient, (b) response of the denitrifying microbial community to physicochemical perturbation over time, and (c) the hypothesized relationship between ecosystem physicochemical variability and the perturbation response.

[Figure]

[Figure]

[Figure]